# Regulation of Serum Sodium Levels during Chemotherapy Using Selective Arginine Vasopressin V2-Receptor Antagonist Tolvaptan in a Four-Year-Old Girl with a Suprasellar Germ Cell Tumor

**DOI:** 10.3390/children8040293

**Published:** 2021-04-10

**Authors:** Shota Hiroshima, Hiromi Nyuzuki, Sunao Sasaki, Yohei Ogawa, Keisuke Nagasaki

**Affiliations:** 1Division of Pediatrics, Department of Homeostatic Regulation and Development, Niigata University Graduate School of Medical and Dental Sciences, Niigata 951-8510, Japan; sho980522@gmail.com (S.H.); nyuzuki@med.niigata-u.ac.jp (H.N.); sunaoenari@gmail.com (S.S.); yohei_oga@yahoo.co.jp (Y.O.); 2Division of Community Medicine, Department of Community Medicine, Niigata University Graduate School of Medical and Dental Sciences, Niigata 951-8510, Japan

**Keywords:** tolvaptan, hyponatremia, pediatric SIADH, germ cell tumor, chemotherapy

## Abstract

There are limited reports on the use of tolvaptan for syndrome of inappropriate antidiuretic hormone secretion (SIADH) in children. Managing serum sodium levels in SIADH patients during chemotherapy is often difficult because of the need for massive fluid infusions. We report the course of the use of tolvaptan for the treatment of hyponatremia during chemotherapy in a four-year-old girl with a suprasellar germ cell tumor. The patient was a Japanese girl who presented with left ptosis with a mass in the pituitary gland and cavernous sinus. She was diagnosed with an intermediate-grade germ cell tumor and was treated with carboplatin and etoposide combination chemotherapy. She developed hyponatremia due to SIADH caused by intravenous infusion therapy before chemotherapy. Subsequently, tolvaptan (3.25 mg; 0.20 mg/kg/dose) was administered orally to control serum sodium levels. After 4 h of administration, a marked increase in urine volume of up to 15 mL/kg/h was observed, and serum sodium level increased from 126 to 138 mEq/L after 10 h of tolvaptan administration, followed by a decrease in urine volume. The use of tolvaptan in pediatric patients with SIADH who require intravenous hydration during chemotherapy can be useful for the management of serum sodium balance.

## 1. Introduction

Syndrome of inappropriate antidiuretic hormone secretion (SIADH) is a condition in which, despite low osmotic pressure, arginine vasopressin (AVP) secretion is not suppressed, AVP is continuously secreted, and excess body fluid accumulates because of the promotion of water reabsorption within the renal collecting tubule, resulting in diluted hyponatremia [1,2]. SIADH can be caused by central nervous system diseases, lung diseases, ectopic vasopressin production, and drugs [3]. The first line of SIADH treatment is to eliminate the underlying cause when possible, and the second line is the restriction of water [3], in addition to hypertonic fluids, loop diuretics, and oral urea [4]. However, managing serum sodium levels in SIADH patients during chemotherapy is often difficult because of the need for massive fluid infusions.

The vasopressin type-2 receptor antagonist tolvaptan can be used to treat hyponatremia. Tolvaptan selectively and competitively binds to and blocks the V2 receptor located in the walls of the vasculature and luminal membranes of renal collecting ducts, thereby preventing the binding of vasopressin to the V2 receptor [1,2]. This prevents water absorption in the renal collecting tubule and increases the excretion of electrolyte-free water via the kidneys [1,2]. This reduces the intravascular volume and increases serum sodium concentration and osmolality. Mozavaptan hydrochloride was previously approved in Japan as a vasopressin type-2 receptor antagonist for the treatment of SIADH in ectopic vasopressin-producing tumors in 2006. In June 2020, tolvaptan was approved in Japan for the treatment of SIADH in adults [5]. Tolvaptan was approved in 2009 in the United States and European countries for the treatment of hyponatremia, but there have been limited reports of its use for SIADH in children.

Herein, we report the course of the use of tolvaptan for the treatment of hyponatremia during chemotherapy in a four-year-old girl with a suprasellar germ cell tumor presenting with SIADH.

## 2. Case Presentation

The patient was the first child of Japanese non-consanguineous parents. She was born at full-term via normal delivery, and her birth weight was 3482 g. Neonatal screening tests were within the normal range. She was a healthy girl until four years of age. She visited our department presenting with left ptosis, left eye abduction failure, and left low vision. Contrast-enhanced T1-weighted imaging revealed a mass with inhomogeneous enhancement in the pituitary region, cavernous sinus, and enlargement of the pituitary stalk. Based on the biopsy results, she was diagnosed with an intermediate-grade germ cell tumor and was treated with carboplatin and etoposide combination chemotherapy (CARE therapy). Serum Na levels at admission and after biopsy were 137 and 132 mEq/L, respectively, ranging from normal to mildly low. Following hydration therapy before chemotherapy, she presented with neck pain and a decrease in serum sodium levels due to SIADH (Table 1). Her cardiac and renal functions were normal, and there were no findings of dehydration. In addition, the insulin, thyrotropin-releasing hormone, and luteinizing hormone-releasing hormone test results showed normal hypothalamic–pituitary–adrenal function.

The course of the first cycle of CARE chemotherapy is shown in Figure 1. A total volume of 1050 mL of fluids was administered on day 1, and her serum sodium levels decreased to 115 mEq/L during CARE therapy on the morning of day 2, and she complained of neck pain and nausea. She was treated for symptomatic hyponatremia with 10 mL of 2.4% NaCl per hour for 4 h by intravenous infusion, and her symptoms improved. Considering the possibility of worsening hyponatremia due to fluid infusion during CARE therapy, we decided to continue CARE therapy using tolvaptan (0.20 mg/kg/dose) after obtaining the mother’s consent. The in-out balance and the serum sodium level every 4 h after tolvaptan use are shown in Figure 2. After 4 h of tolvaptan administration, urine volume increased remarkably (maximum 15 mL/kg/h). After 10 h of tolvaptan administration, the serum sodium level was normal (138 mEq/L). After 12 h, the urine volume decreased prior to tolvaptan treatment. On the third day, tolvaptan (0.20 mg/kg/dose) was again administered to prevent hyponatremia, and the serum sodium level remained normal during treatment (Figure 1). Subsequently, hyponatremia was also observed during the second and third courses of CARE therapy, and tolvaptan was used to appropriately manage the serum sodium levels.

This was a retrospective analysis of the clinical course of a patient examined and treated at the Niigata University Medical and Dental Hospital in Japan. The use of tolvaptan in this case was approved by the Clinical Ethics Review Committee of Niigata University Medical and Dental Hospital. Written informed consent was obtained from the mother.

The upper panel shows the changes in serum sodium levels and urine-specific gravity. The lower panel shows the sodium content of intravenous infusion. The color intensity of the square is proportional to the concentration of Na.

## 3. Discussion

The cornerstone of SIADH treatment is water restriction, but fluid balance can be difficult to manage when fluids are needed, such as during chemotherapy. We report a pediatric case of hyponatremia due to hydration during chemotherapy in a four-year-old girl with a suprasellar germ cell tumor.

Reports of tolvaptan use for SIADH in children are limited [6,7,8,9,10,11]. The reported pediatric patients are summarized in Table 2. The age range was 0.2 to 17 years, and tolvaptan was used at various ages throughout childhood. Intracranial tumors were the most common cause of SIADH, but idiopathic and traumatic patients were also present. The initial dose of tolvaptan ranged from 0.05 to 0.8 mg/kg/dose. Referring to the initial dose of adult SIADH [5] and the report by Willemsen et al. [6] we have chosen an initial dose of 0.2 mg/kg/dose with good results.

Management of sodium balance for SIADH during chemotherapy with massive infusion is difficult. Willemsen et al. reported the use of tolvaptan for hyponatremia associated with massive infusion during methotrexate therapy for intracranial B-cell lymphoma [6]. Tolvaptan treatment is useful in the management of serum sodium levels through a gradual increase in sodium levels, allowing for liberalizing fluid intake and hyperhydration [6]. In this study, we used tolvaptan prophylactically in combination with massive infusion and were able to keep the serum sodium level almost constant.

The onset and duration of action of tolvaptan in pediatric patients with SIADH are unknown. In the US Guidelines for hyponatremia, it is essential that the serum sodium concentration is measured frequently during the active phase of correction of hyponatremia at a minimum of every 6 to 8 h during the use of tolvaptan in adult patients with SIADH, particularly in patients with risk factors for osmotic demyelination [12]. In this study, the onset time of tolvaptan effect was 1 to 2 h, and the maximum effect time was 4 to 8 h, lasting approximately 12 h, and serum sodium increased by 12 mEq/L in 10 h. As data on the use of tolvaptan in children are scarce and safety cannot be guaranteed, closer monitoring is required during the use of tolvaptan in children than that in adults.

## 4. Conclusions

The use of tolvaptan in pediatric patients with SIADH who require intravenous hydration during chemotherapy can be useful for the management of serum sodium balance.

## Figures and Tables

**Figure 1 children-08-00293-f001:**
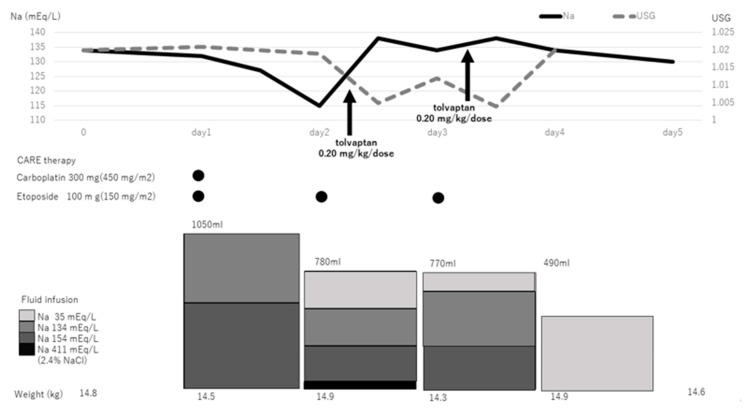
The first cycle of CARE chemotherapy. USG, urine specific gravity; CARE, carboplatin and etoposide combination chemotherapy.

**Figure 2 children-08-00293-f002:**
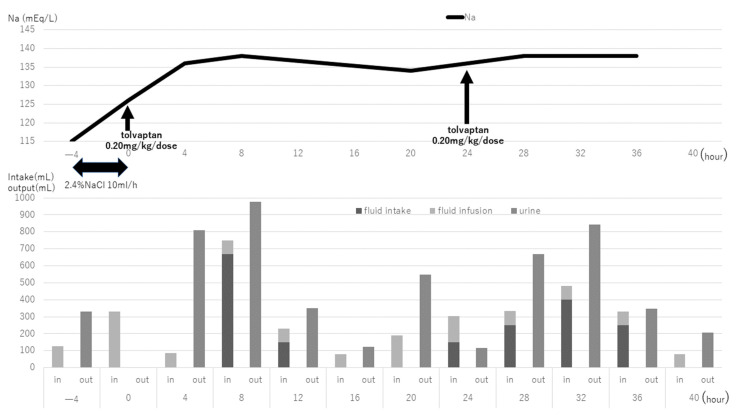
Changes in serum sodium levels and water intake and urine output after use of tolvaptan. The data in the bottom panel show the in-out balance data every 4 h.

**Table 1 children-08-00293-t001:** Laboratory data in hyponatremia before the chemotherapy.

Blood Chemistry			Endocrine Tests			(Reference Values)
TP	6.5	g/dL	ADH	1.1	pg/mL	<0.5
BUN	8.0	mg/dL	BNP	<5.8	pg/mL	<18.4
UA	3.9	mg/dL	Aldosterone	6.0	ng/mL	3.6–24.0
Cre	0.19	mg/dL	PRA	1.0	ng/mL/h	0.2–2.7
Na	121	mEq/L	ACTH	8.9	pg/mL	7.2–21.0
K	3.6	mEq/L	Cortisol	5.5	μg/dL	6.4–21.0
Cl	93	mEq/L				
TG	198	mg/dL				
BG	106	mg/dL				
Plasma Osm	252	mOsm/kg				
Urinalysis						
Urine specific gravity	1.024					
Na	194	mEq/L				
Cre	48	mEq/L				
Urine Osm	708	mOsm/kg				

TP, total protein; BUN, blood urea nitrogen; UA, uric acid; TG, triglycerides; BG, blood glucose; Osm, osmolality; Cre, creatinine; ADH, antidiuretic hormone; BNP, brain natriuretic peptide; PRA, plasma renin activity; ACTH, adrenocorticotropic hormone.

**Table 2 children-08-00293-t002:** Reported pediatric patients that were administered tolvaptan for SIADH.

Authors	Age (Years)	Cause of SIADH	Initial Dose of Tolvaptan (mg/kg/dose)	Change in s-Na (mEq/L) *	Maximum UV (mL/kg/h) *	References
Koksoy, et al.	16	Idiopathic	0.28	115 to 122 in 16 h	9	[10]
Marx-Berger, et al.	0.2	Idiopathic	0.8	138 to 153 in 2 days	No data	[9]
Marx-Berger, et al.	0.3	Idiopathic	0.6	No data	No data	[9]
Willemsen, et al.	11	Intracranial B-cell lymphoma	0.14	119 to 130 in 2 days	8	[6]
Tuli, et al.	7	ROHHAD syndrome	0.06	No data	No data	[7]
Tuli, et al.	4	A large sellar and suprasellar tumor	0.1	No data	No data	[7]
Tuli, et al.	5	Hypothalamic astrocytoma	0.05	No data	No data	[7]
Gürbüz, et al.	13	Craniopharyngioma	0.13	117 to 126 in 4 h	8.1	[8]
Kraayvanger, et al.	17	Severe polytrauma	15 mg/day	116 to 139 in 24 h	No data	[11]
Hiroshima, et al.	4	Suprasellar germ cell tumor	0.2	126 to 138 mEq/L in 10 h	15 at 4 h	This study

* Changes after administration of tolvaptan; s-Na, serum sodium level; Osm, plasma osmolality; UV, urine volume; ROHHAD, rapid-onset obesity with hypothalamic dysregulation.

## Data Availability

The data presented in this study are available upon request from the corresponding author. The data are not publicly available due to privacy restrictions.

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
