# Peer review of "Regulation of Serum Sodium Levels during Chemotherapy Using Selective Arginine Vasopressin V2-Receptor Antagonist Tolvaptan in a Four-Year-Old Girl with a Suprasellar Germ Cell Tumor"

_children, 2021, doi:10.3390/children8040293_

Round 1

Reviewer 1 Report

Interesting case report.

Introduction: line 31 "is not suppressed "(to change)

The authors must also mention the treatment with urea in SIADH or NSIAD in children (Vandergheynst et al,Nephron,etc).I agree that vaptans is a better choice if a transient high diuresis is needed like needed with some chemotherapy.In most cases of NSIAD ,diuresis does not increase with tolvaptan...

Author Response

We wish to express our appreciation to the reviewer for his or her insightful comments, which have helped us significantly improve the paper.

Comment 1: Introduction: line 31 "is not suppressed "(to change)

Response: We have changed line 31 "is not suppressed "

Comment 2: The authors must also mention the treatment with urea in SIADH or NSIAD in children (Vandergheynst et al,Nephron,etc).I agree that vaptans is a better choice if a transient high diuresis is needed like needed with some chemotherapy.In most cases of NSIAD ,diuresis does not increase with tolvaptan...

Response: We have added the following text in line 37 “in addition to hypertonic fluids, loop diuretics and oral urea [4]." 

Thank you again for your comments on our paper. We trust that the revised manuscript is suitable for publication.

Reviewer 2 Report

Comments to the Authors
The authors present a well written manuscript.  The study describes a case of tolvaptan therapy of SIADH in a girl with diagnosis of brain tumor. There are limited data about the use of vasopressin receptor 2 antagonist in children with SIADH.

Comments:

Introduction:

Among the possible treatment options for SIADH, besides the fluid restriction, is hypertonic saline infusion and diuretics. It should be added to the text.

Case presentation:

  1. Thera are no data about initial serum sodium level (at admission and after biopsy of the tumor). In my opinion those data are very important, as hyponatremia could be also related to pituitary surgery or tumor itself.
  2. How was the general condition of the patient at hyponatremia onset and during treatment? Was the hyponatremia asymptomatic?
  3. In table 1 – column with normal values of presented parameters would help to analyze the data.

Discussion:

There is a wide range of tolvaptan dose used in children presented in literature (Table 2). Why the authors decided to give a dose 0,20 mg/kg/dose? Please explain this decision in discussion.

Author Response

We wish to express our appreciation to the reviewer for his or her insightful comments, which have helped us significantly improve the paper.  

Comment 1: Among the possible treatment options for SIADH, besides the fluid restriction, is hypertonic saline infusion and diuretics. It should be added to the text. Response: We have added the following text in line 37 "in addition to hypertonic fluids, loop diuretics and oral urea [4]."  

Comment 2: Thera are no data about initial serum sodium level (at admission and after biopsy of the tumor). In my opinion those data are very important, as hyponatremia could be also related to pituitary surgery or tumor itself. Response: We have added serum Na levels on admission and after surgery.

Comment 3: How was the general condition of the patient at hyponatremia onset and during treatment? Was the hyponatremia asymptomatic?

Response: At the onset of hyponatremia, she complained of neck pain. We have added text for this content in Line 66.  

Comment 4: In table 1 – column with normal values of presented parameters would help to analyze the data.

Response: We have added reference values, mainly for endocrine tests.   Comment 5: There is a wide range of tolvaptan dose used in children presented in literature (Table 2). Why the authors decided to give a dose 0,20 mg/kg/dose? Please explain this decision in discussion.

Response: We have added the following text in line 112-114 " Referring to the initial dose of adult SIADH [5] and the report by Willemsen et al. [6] we have chosen an initial dose of 0.2 mg/kg/dose with good results."  

Thank you again for your comments on our paper. We trust that the revised manuscript is suitable for publication.